# Behavior of Concrete Reinforced with Date Palm Fibers

**DOI:** 10.3390/ma15227923

**Published:** 2022-11-09

**Authors:** Fadi Althoey, Ibrahim Y. Hakeem, Md. Akter Hosen, Shaker Qaidi, Haytham F. Isleem, Haitham Hadidi, Kiran Shahapurkar, Jawad Ahmad, Elias Ali

**Affiliations:** 1Civil Engineering Department, Najran University, Najran P.O. Box 1988, Saudi Arabia; 2Department of Civil and Environmental Engineering, Dhofar University, Salalah P.O. Box 2509, Oman; 3Department of Civil Engineering, College of Engineering, University of Duhok, Duhok 42001, Iraq; 4Department of Civil Engineering, College of Engineering, Nawroz University, Duhok 42001, Iraq; 5Civil Engineering Department, Qujing Normal University, Qujing 655011, China; 6Department of Mechanical Engineering, College of Engineering, Jazan University, Jazan P.O. Box 114, Saudi Arabia; 7Department of Mechanical Engineering, School of Mechanical, Chemical and Materials Engineering, Adama Science and Technology University, Adama 1888, Ethiopia; 8Department of Civil Engineering, Military College of Engineering (NUST), Resulpur 24080, Pakistan; 9Department of Civil and Environmental Engineering, Case Western Reserve University, Cleveland, OH 44106, USA

**Keywords:** sustainability, date palm fibers, high-strength concrete, engineering characteristics

## Abstract

In recent decades, researchers have begun to investigate innovative sustainable construction materials for the development of greener and more environmentally friendly infrastructures. The main purpose of this article is to investigate the possibility of employing date palm tree waste as a natural fiber alternative for conventional steel and polypropylene fibers (PPFs) in concrete. Date palm fibers are a common agricultural waste in Middle Eastern nations, particularly Saudi Arabia. As a result, this research examined the engineering properties of high-strength concrete using date palm fibers, as well as the performance of traditional steel and PPF concrete. The concrete samples were made using 0.0%, 0.20%, 0.60%, and 1.0% by volume of date palm, steel, and polypropylene fibers. Ten concrete mixtures were made in total. Compressive strength, flexural strength, splitting tensile strength, density, ultrasonic pulse velocity (UPV), water absorption capability, and water permeability tests were performed on the fibrous-reinforced high-strength concrete. With a 1% proportion of date palm, steel, and polypropylene fibers, the splitting tensile strength improved by 17%, 43%, and 16%, respectively. By adding 1% fiber, flexural strength was increased by 60% to 85%, 67% to 165%, and 61% to 79%. In addition, date palm fibers outperformed steel and PPFs in terms of density, UPV, and water permeability. As a result, date palm fibers might potentially be employed in the present construction sector to improve the serviceability of structural elements.

## 1. Introduction

Environmental pollution is a crucial cause of climate change that may lead to a risk to human health. Hence, the efficient management of waste materials is essential for a smooth industrial revolution [1,2]. Researchers have recently concentrated on waste materials by converting them into wealth, such as energy production [3,4,5]. Further, waste materials might be reprocessed and reused to advance other valuable materials [6,7,8]. Infrastructure construction materials, particularly concrete, have been integrated with waste materials as critical components for several years [9,10,11]. Numerous studies have incorporated waste materials into concrete, such as gravel aggregates substituted by plastics [12], rubber [13], bottle glass [14], and recycled concrete aggregates [15]. Applying natural fibers as a construction material requires a unique challenge in the modern construction industry. Natural fibers have encountered the task of emerging new methods to simplify the usages of natural fibers in the fabrication of construction materials, where their benefits consent to compete efficiently [16,17,18]. The extraordinary challenges for the researcher are fiber quality, orientation and distribution, fiber strength, fiber humidity, fiber hydrophilicity, concerns of compatibility, and fiber degradation during chemical processing [19]. The application of natural fibers is more advantageous as fibers are locally accessible in abundance, are cost-effective, have small energy utilization during the process and permit a decrease in impacts on the environment. Aslam et al. observed that the addition of 2% coconut fibers in waste glass concrete improved the density by 20% and enhanced the mechanical strength [20].

Palm fibers have strand textures with distinct properties such as being low-cost, available in some regions, lightweight, and durable [21]. Fibers extracted from rotten palm trees are found to have low tensile strength, brittle behavior, low elasticity modulus, and very high water absorption capacity [22]. Several studies have incorporated natural fibers into construction materials [23,24,25,26,27]. For instance, Minke [28] observed that the addition of natural fibers into clay materials such as coir, animal or human hair, agave, sisal, straw, and bamboo might help in decreasing the shrinkage due to the dilution of the clay content as well as the definite amount of water immersed by the pores of the fiber. Taallah, Bachir, et al. [29] investigated the mechanical and hygroscopicity characteristics of compressed earth blocks occupied by date palm fibers to ensure the use of local building materials and support for low-cost housing in rural areas. Rokbi M. et al. [30] investigated developing and characterizing a polymer-reinforced concrete with date palm fibers. Their study used three sizes (short, very short, or mixed) of date palm fibers as a reinforcement. The experimental outcomes showed that the compressive and flexural strength were improved in some specimens. The study concluded that the strength improvement or degradation was attributed to the nature (treated or untreated) and the fibers’ size (short, very short, or mixed).

Since a lack of information on the engineering characteristics of the agro-waste date palm fiber-reinforced high-strength concrete found in the literature, the objective of this study is to examine the engineering characteristics of the date palm fiber-reinforced concrete in all aspects and make comparisons with conventional steel and polypropylene fibrous concrete.

## 2. Materials and Specimens Preparation

### 2.1. Materials

#### 2.1.1. Cement

This study used ordinary Portland cement Type-I for fabricating the high-strength date palm, steel, and polypropylene fibrous concrete specimens. The fineness and specific gravity of the cement were 410 m^2^/kg and 3.15, respectively. The cement contained 59% C_3_S, 12.10% C_2_S, 10.60% C_3_A, and 10.4% C_4_AF, as reported by the manufacturer [31] and confirmed with ASTM C 150 [32]. The chemical composition of the cement is recorded in Table 1.

#### 2.1.2. Aggregates

The fine aggregate consists of natural dune sand with maximum particles passing through sieve No.4. More information about the fine aggregate used herein can be seen in previous work by the authors [31]. The coarser aggregate was a crushed stone with a maximum size of 20 mm used for manufacturing the high-strength date palm, steel, and polypropylene fibrous concrete. The physical characteristics of the fine and coarse aggregates are exhibited in Table 2.

#### 2.1.3. Water and Superplasticizer

The filtered tap water was utilized as a fundamental element in manufacturing the high-strength date palm, steel, and polypropylene fibrous concrete and curing concrete. The properties of the water have accomplished the requirement of fabricating high-strength fibrous concrete with ASTM C1602/C1602M [33]. Super Plasticizers (SP) are well-known as great water reducers for manufacturing high-strength fibrous concrete. Sika Viscocrete, constructed on polycarboxylate ether (PCE), was employed as an SP for water reducers in this study.

#### 2.1.4. Date Palm Fibers

The date palm fibers were obtained from around Najran city, Saudi Arabia’s date palm trees are aged between 15 and 25 years old, which signifies one of the best available diversities and are accountable for significant agricultural waste production. These date palm fibers were manually collected from the palm trees. The date palm fibers are sited around the tree’s trunk in a bidirectional form, comprised of two or three layers packed and superimposed. The collected raw date palm fibers had different diameters and lengths. Figure 1 shows the process of collecting the raw date palm material and turning it into fibers. The date palm fibers were cured chemically with 3% concentrations of analytical Sodium Hydroxide to eliminate any possible impurities from the surfaces of the fibers, and this chemical treatment also enhanced their matrix compatibility [34]. Afterward, the fibers were immersed in sulfuric acid (0.5% concentration) and washed with deionized (DI) water. Finally, the fibers were dried by sunlight and manually cut into 60 mm lengths with varying diameters. The physical properties of date palm fibers are revealed in Table 3.

#### 2.1.5. Steel Fibers

The steel fibers were hooked at both ends, which were bundles glued as demonstrated in Figure 2. This glued type of fiber was used for fabricating high-strength fibrous concrete. The physical properties of steel fibers are presented in Table 4. 

#### 2.1.6. Polypropylene Fibers

This study used polypropylene fiber (Figure 3) to fabricate the high-strength fibrous concrete compared with date palm and fibrous steel concrete. The manufacturer delivered the Polypropylene fiber’s physical properties, as shown in Table 5.

### 2.2. Specimens Preparation

Using different fractions (0%, 0.20%, 0.60%, and 1.0%) by binder weight for date palm, steel, and polypropylene fibers, a total of 10 mixtures were fabricated. The experimental program was carried out on 100 mm cubes, 150 mm diameter × 300 mm height cylinders and 100 × 100 × 500 mm^3^ prisms to compute different hardened characteristics of the high-strength palm fibrous concrete. The configurations of the high-strength fibrous concrete mixtures are revealed in Table 6.

## 3. Testing Methods

### 3.1. Compressive Strength Test

The high-strength date palm, steel, and polypropylene fibrous concrete specimens were tested according to BS EN12390-3 [36] under the compression load utilizing a 2000 kN capability of a mechanical compression testing machine (MATEST) with a loading rate of 0.0167 kN/sec. The size of the cube specimens was 50 × 50 × 50 mm^3^. The specimens were tested after the 28-day curing regimen. Prior to testing, the specimens were kept at ambient temperature (23 °C ± 2 °C) for 24 h. Three specimens were tested for each mixture, and the average values were reported.

### 3.2. Flexural Strength Test

The flexural strength of the high-strength date palm, steel, and polypropylene fibrous concrete specimens was executed under a four-point loading arrangement that was implemented as per the BS EN 12390-5 [37], which was applied through a Universal Instron machine (400 kN loading capacity) with a constant loading rate of 1.0 mm/min. This experiment also assessed the modulus of rupture (MOR) of the high-strength date palm, steel, and polypropylene fibrous concrete. This technique is prevalent for accomplishing the flexural strength of high-strength fibrous concrete. The displacement of the 100 × 100 × 500 mm^3^ high-strength fibrous concrete prism specimens was computed utilizing a Linear variable displacement transducer (LVDT) attached to the center of the prism specimens. The applied load and displacement were automatically recorded in the TDS-530 data logger during the execution of the experiment on the samples. The recorded results were transferred from the data logger to a computer to analyze the load-displacement graphs.

### 3.3. Splitting Tensile Strength Test

The splitting tensile strengths of the high-strength date palm, steel, and polypropylene fibrous concrete were assessed at 28-days. The cylindrical high-strength fibrous concrete specimens were used for splitting tensile strength. The cylindrical specimens with dimensions of 100 mm diameter and 200 mm height and experiments were executed following BS EN 12390-6 [38].

### 3.4. Density Test

The density of the high-strength date palm, steel, and polypropylene fibrous concrete specimens was measured after the curing period of 28-days. This experiment was carried out on the samples before the mechanical compression test. These experiments contained the determination of the weight and volume of the fibrous concrete specimens. The density of the fibrous concrete samples is determined, conferring to the formula: density = weight/volume.

### 3.5. Ultrasound Plus Velocity Test

The integrity and homogeneity of the high-strength date palm, steel, and polypropylene fibrous concrete specimens were confirmed by the ultrasonic pulse velocity (UPV) test [39,40]. The test was conducted on the fibrous concrete samples following the ASTM C597 [41].

### 3.6. Water Absorption Capacity Test

The superior quality of concrete is labeled by encompassing trivial pores changed by excessive water. Therefore, the water absorption capacity measuring experiment is usually applied to compute concrete quality, such as density, rigidity, and durability. The water absorption test for high-strength date palm, steel, and polypropylene fibrous concrete was conducted by conferring to BS 2011 Part 122 [42] having the cylindrical specimens with sizes of 75 mm in diameter and 150 mm in height after achieving the requirement of the curing period of 28 days. Initially, the fibrous concrete samples were dried in an electric power oven for 72 h, maintaining a temperature of 105 °C. Then, the specimens were collected from the oven, cooled in a dry airtight container for 24 h, and weighed. The samples were immediately immersed in a water tank encompassing 20 °C temperature. The specimen’s longitudinal axis was preserved horizontally, with a water depth of 25 mm on the samples for 30 min. Finally, the samples were collected from the water, the clothes were dried to achieve the saturated surface condition, and they were reweighed. The water absorption capacity of fibrous concrete specimens was determined as the increase in the weight owing to immersion under water, which was revealed as a percentage of the dry weight of the sample.

### 3.7. Water Permeability Test

The water penetration intensity into the high-strength date palm, steel, and polypropylene fibrous concrete specimens was evaluated using a Zwick machine [43]. This experiment was carried out using three 150 mm cubic fibrous concrete samples after 28 days. The specimens were sustained at a constant water pressure of 0.50 N/mm^2^ for an irreparable period of 72 h. After the specified time, the fibrous concrete samples were removed from water pressure and divided into two halves, while the water-exposed specimen’s face was shown to be face down. The intensity of water penetration into the fibrous concrete samples was recorded and assessed as the permeability.

## 4. Experimental Outcomes and Discussions

### 4.1. Compressive Strength

The compressive strength of date palm, steel, and polypropylene fibrous-reinforced high-strength concrete with different volume fractions of fibers is revealed in Figure 4a. The compressive strength of the high-strength fibrous concrete primarily depended on the coarse aggregate’s strength [25,44,45]. Hence, there was an insignificant impact of the fibers on the compressive strength of the high-strength fibrous-reinforced concretes assessed. Including date palm, steel, and polypropylene fibers of 1.00% into the concrete increased the utmost compressive strength by 8.01%, 9.60%, and 7.53%, respectively, over the reference concrete specimens at 28 days. Whereas 0.20% and 0.60% of date palm, steel, and polypropylene fibers content did not exhibit any noticeable influence on the high-strength concrete compressive strength, which might be owing to the deficiency of fiber quantity in the concrete. Alatshan, F. et al. [46] revealed that the maximum compressive strength was achieved at 30 MPa with 1.0% of date fibers in the concrete. Figure 4b illustrates the faces of the specimen during the execution of loading, showing its superior load-bearing proficiency and ductility since the compressive strength of specimens was notably enhanced by the fibers.

The correlation between the compressive strength and the addition of date palm, steel, and polypropylene fibers to the high-strength concrete is revealed in Figure 5. Therefore, the following equations are specified based on the relationships between the compressive strength and the addition of different fibers.
(1)$$f_{cd}=7.57V_{fd}+62.51$$
(2)$$f_{cs}=5.99V_{fs}+63.01$$
(3)$$f_{cp}=3.32V_{fp}+65.54$$
where *f_cd_*, *f_cs_*, and *f_cp_* are the compressive strengths of the date palm, steel, and polypropylene fibrous-reinforced high-strength concrete, respectively, and *V_fd_*, *V_fs_* and *V_fp_* are the volume fractions of date palm, steel, and polypropylene fibers, respectively.

### 4.2. Flexural Strength

The flexural strength and impact resistance of fiber-reinforced concrete is very significant when infrastructures are exposed to the marine environment or extreme weathering conditions [47,48]. The flexural strength of high-strength concrete and its enhancement comprising 0.20%, 0.60%, and 1.0% date palm, steel, and polypropylene fibers at 28 days of curing are demonstrated in Figure 6. The significant gradual enhancement of flexural strength with date palm and steel fibers was remarked from 60% to 85% and 67% to 165% compared with the reference specimen. At the same time, the polypropylene fibers were improved from 61% to 79%. Hence, it was noticed that flexural strength enhancement of fibrous concrete predominantly depends on the tensile strength of the fibers because the steel fibers are better than date palm and polypropylene fibers to counterattack or delay the formation of primary cracks. It was observed that all specimens failed by flexure. It is worth mentioning that the flexural strength of concrete comprising different natural materials (e.g., kelp, coconut, and jute fibers) was improved by up to 50% with respect to the control specimen [44].

The correlation between the flexural strength and volume fractions of the date palm, steel, and polypropylene fibrous concrete is represented in Figure 7. As a result, the following equations are suggested for predicting the date palm, steel, and polypropylene fibrous concrete flexural strength based on the outcomes from experimental investigations.
(4)$$f_{rd}=9.62V_{fd}^{0.09}$$
(5)$$f_{rs}=13.27V_{fs}^{0.28}$$
(6)$$f_{rp}=9.39V_{fp}^{0.07}$$
where *f_rd_*, *f_rs_* and *f_rp_* is the flexural strength of the date palm, steel, and polypropylene fibrous-reinforced high-strength concrete, respectively.

### 4.3. Splitting Tensile Strength

The splitting tensile strength characteristics of the date palm, steel, and polypropylene fibrous high-strength concrete are exhibited in Figure 8. The maximum enhancement of up to 43% of splitting tensile strength was achieved with steel fibers. In contrast, date palm and polypropylene fibers increased by up to 17% and 16%, respectively, compared with the reference specimens using 1.0% of fibers in the concrete. This might occur due to the stiffness and roughness of the surface of the fibers. On the other hand, the fewer volume fractions of fibers (0.20% and 0.60%) in the high-strength concrete did not have an efficient impact on the fibrous concrete, especially for date palm and polypropylene fibers.

Figure 9 exhibits the correlation between the splitting strength and volume fractions of date palm, steel, and PPFs. The following equations might be beneficial for predicting the date palm, steel, and polypropylene fibrous high-strength concrete splitting tensile strength, which was established based on the outcomes of the experimental investigations.
(7)$$f_{td}=6.82V_{fd}^{0.09}$$
(8)$$f_{ts}=8.55V_{fs}^{0.13}$$
(9)$$f_{tp}=6.71V_{fp}^{0.07}$$
where *f_td_*, *f_ts_*, and *f_tp_* is the splitting tensile strength of the date palm, steel, and PPFs reinforced high-strength concrete, respectively.

### 4.4. Density of Concrete

The densities of the high-strength fibrous concrete incorporating 0%, 0.20%, 0.60%, and 1.0% of date palm, steel, and polypropylene fibers at 28 days are demonstrated in Figure 10. The high-strength concrete encompassing date palm and polypropylene fibers shows that increasing the volume fractions of fibers reduces the densities of the concrete concerning the reference specimens. These concretes with a maximum 1.0% volume fraction of date palm and polypropylene fibers reduced the densities by 1% and 2%, respectively, compared to the reference specimens. On the other hand, increasing the steel fiber volume fraction up to 1.0% in the high-strength concrete intensified the density by 3% compared to reference specimens. It might be concluded that the increment or decline of the high-strength concrete’s densities depends on the fibers’ weight and volume fractions.

The correlation between the density of high-strength concrete and volume fractions of date palm, steel, and polypropylene fibers is represented in Figure 11. The figure revealed that densities of the date palm and polypropylene fibrous-reinforced high-strength concrete linearly decreased with the increasing volume fractions of the fibers. Contrastingly, steel fibrous-reinforced high-strength concrete progressively increases with adding fibers to the concrete. The densities of the high-strength date palm, steel, and polypropylene fibrous concrete might be predicted utilizing the following equations.
(10)$$\gamma_{d}=-22.71V_{fd}+2520.41$$
(11)$$\gamma_{s}=\;57.75V_{fs}+2528.52$$
(12)$$\gamma_{p}=-52.25V_{fp}+2514.18$$
where *γ_d_*, *γ_s_*, and *γ_p_* are the density of the high-strength date palm, steel, and PPFs, respectively.

Numerous studies have revealed that concrete density increases by increasing steel fiber’s contents in the concrete [45,49]. Moreover, the density of the concrete is reduced by increasing the polypropylene fibers in the concrete [46,47].

### 4.5. Ultrasound plus Velocity (UPV)

The relationship between the UPV and volume fraction of the date palm, steel, and polypropylene fibers in the concrete is presented in Figure 12. The UPV of the high-strength fibrous concrete demonstrated a descending trend with increasing volume fractions of date palm, steel, and polypropylene fibers. Hence, the concrete’s date palm, steel, and polypropylene fibers increased the porosity and decreased the density. Meanwhile, the UPV of the concrete is an execution of the volumetric intensity of ingredients in the concrete [48]. The specimens comprising steel fibers revealed the highest UPV drop compared with samples containing date palm and polypropylene fibers. Therefore, the date palm and polypropylene fibrous-reinforced high-strength concrete were more homogeneous and less porous than steel fibrous-reinforced concrete.

### 4.6. Water Absorption Capability

The water absorption capability for date palm, steel, and polypropylene fibrous-reinforced high-strength concrete is shown in Figure 13. Since the steel fibers could easily flow around the aggregates and the fibers construct a strong bridge with them, therefore, steel fibers diminished the micro-cracks by interconnecting each other and enriching the microstructural characteristics of the concrete. Consequently, the steel fibers in the high-strength concrete significantly reduced the water absorption capability; contrastingly, the date palm and polypropylene fibrous concrete increased up to 6% and 27%, respectively, over the reference specimen. Neville, A.M. [49] specified that the water absorption capability should be less than 10% for decent-quality concrete.

### 4.7. Water Permeability

Figure 14 illustrates the variation in the water permeability as a function of the date palm, steel, and polypropylene fibers content in the concrete. Figure 14 shows that the high-strength fibrous concrete’s water permeability increased with the fibers’ increasing volume fraction. The water permeability for high-strength date palm fibrous concrete was increased by 14%, 96%, and 143% by the volume fractions of fibers 0.20%, 0.60%, and 1.0%, respectively, over the reference specimen. The fibrous steel concrete was increased by −29%, −9%, and 25% by the volume fractions of fibers 0.20%, 0.60%, and 1.0%, respectively, compared to the reference specimen. On the other hand, the polypropylene fibrous concrete was increased by −11%, 5%, and 16% by the volume fractions of fibers 0.20%, 0.60%, and 1.0%, respectively, compared to the reference specimen. The lowest increment of water permeability was observed for the fibrous steel-concrete due to the physio-chemical characteristics of the steel fibers, which did not allow the penetration of the water into the concrete compared with date palm and polypropylene fibers. Hence, the date palm and polypropylene fibers build a debilitated interfacial bonding, which results in concrete permitting the development of a bigger pore network than steel fibers concrete.

The relationship between the water permeability of high-strength concrete and volume fractions of date palm, steel, and polypropylene fibers is shown in Figure 15. The curves have revealed that the water permeability of the date palm fibrous-reinforced high-strength concrete steeply increases with the fibers’ volume fractions. Contrastingly, steel and polypropylene fibrous-reinforced high-strength concrete follow the slowly growing trend of adding fibers to concrete. The following equations might predict the water permeability of the high-strength date palm, steel, and polypropylene fibrous concrete.
(13)$$\mu_{d}=7.5V_{fd}+4.11$$
(14)$$\mu_{s}=3.13V_{fs}+2.60$$
(15)$$\mu_{p}=1.57V_{fp}+3.90$$

*Μd*, *µs*, and *µp* are the water permeability of the high-strength date palm, steel, and polypropylene fibrous concrete, respectively.

### 4.8. Load-Displacement Characteristics

The load-displacement characteristics of fiber-reinforced high-strength concrete are very important as it implies the serviceability of structural elements. The load-displacement relationship is usually affected by the initial cracking, yield and ultimate loads, and stiffness of the elements. Figure 16 shows the load-displacement curves of the fiber-reinforced high-strength concrete specimens. The curves have shown a nearly identical trend for all specimens except S-0.6 and S-1.0 specimens due to the substantial impact of the steel fibers. Since steel fibers are more ductile and have higher flexural rigidity than date palm and polypropylene fibers, the elastic stage was more dominant than the plastic phase of the fiber-reinforced high-strength concrete specimens.

### 4.9. Ductility

Ductility is expressed as the competence of a material or infrastructure to tolerate substantial plastic deformation without significant loss in strength proficiency [50]. Hence, ductility is an essential characteristic since it allows stress rearrangement and permits safeguards to be considered in the case of approaching failure [51,52,53,54,55]. The ductility index of the fibrous-reinforced high-strength concrete specimens was evaluated from the load versus displacement curves (Figure 16). The ductility index of the specimens is distinct as the relationship of displacement at the extreme load to yield load [56,57,58,59,60]. The ductility and its improvement in the fiber-reinforced high-strength concrete specimens are explicitly revealed in Figure 17. The ductility index of the date palm, steel, and polypropylene fiber-reinforced concrete specimens was enhanced by up to 129% compared with the reference specimen. For the date palm and polypropylene fiber-reinforced concrete specimens’ ductility gradually increased, whereas steel fiber-reinforced concrete specimens’ ductility significantly dropped after 0.60% of steel fiber. Therefore, the date palm fiber-based concrete should be very competent in terms of ductility [61,62,63,64,65,66].

## 5. Conclusions

This study examined the distinct engineering characteristics of high-strength concrete incorporating agro-waste date palm fibers and compared it with conventional steel and polypropylene fibrous concrete. Hence, this experimental investigation emphasized the compressive strength, flexural strength, splitting tensile strength, density, ultrasonic pulse velocity (UPV), water absorption capability, and water permeability of the date palm, steel, and polypropylene fiber-reinforced high-strength concrete. The following conclusions can be drawn from this study.

The impact of steel fibers on the compressive strength was superior compared with the date palm and PPF-reinforced concrete due to the better interconnecting bonds between the fibers and concrete matrix.The flexural strength of the fibrous-reinforced high-strength concrete was significantly improved due to the addition of date palm, steel, and polypropylene fibers. Compared with the reference specimens, the flexural strength was enhanced by about 60% to 85%, 67% to 165%, and 61% to 79% with fibers.The splitting tensile strength of the fibrous-reinforced high-strength concrete was substantially enhanced with the volume fractions of date palm, steel, and polypropylene fibers. The splitting tensile strength was increased by 17%, 43%, and 16% when the percentage of date palm, steel, and polypropylene fibers was 1.0% compared to the reference specimen since the tensile strength of steel fibers was much higher than the date palm and polypropylene fibers.The density of the fibrous-reinforced high-strength concrete improved to 2514.50 to 2496.33 kg/m^3^, 2543.20 to 2589.40 kg/m^3^, and 2502.40 to 2460.60 kg/m^3^ with the addition of fibers.The UPV of the fibrous-reinforced high-strength concrete revealed a declining trend with increasing volume fractions of date palm, steel, and polypropylene fibers.The smallest increase in water permeability was remarked for the steel fibers compared with date palm and polypropylene fibers because the steel fiber’s physio-chemical characteristics did not permit water penetration into the concrete.The ductility of the fibrous-reinforced high-strength concrete increased by up to 129% over the reference specimen. This enhancement was correlated with increasing the fiber volume in the concrete.

## Figures and Tables

**Figure 1 materials-15-07923-f001:**
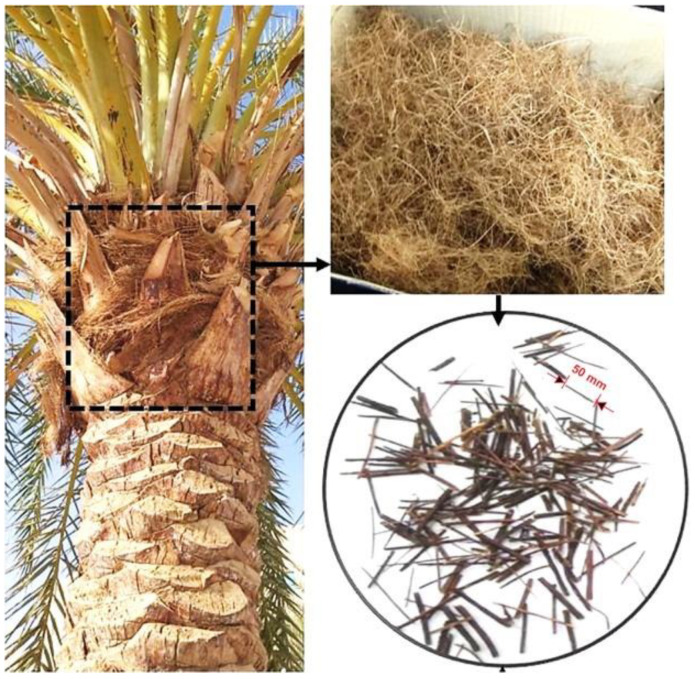
Date palm fibers as an agro-waste material.

**Figure 2 materials-15-07923-f002:**
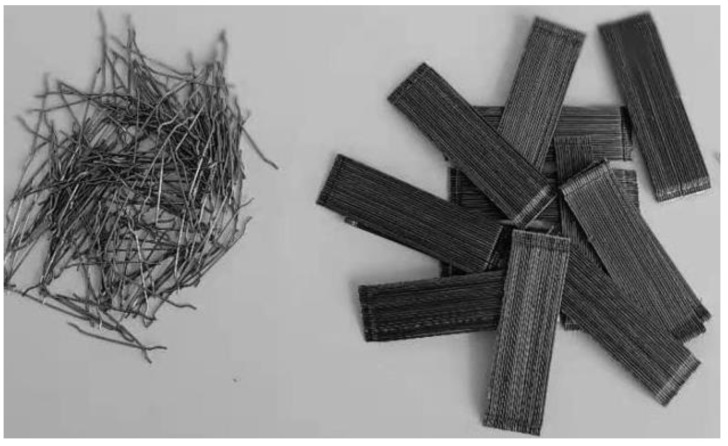
Glued Steel fibers.

**Figure 3 materials-15-07923-f003:**
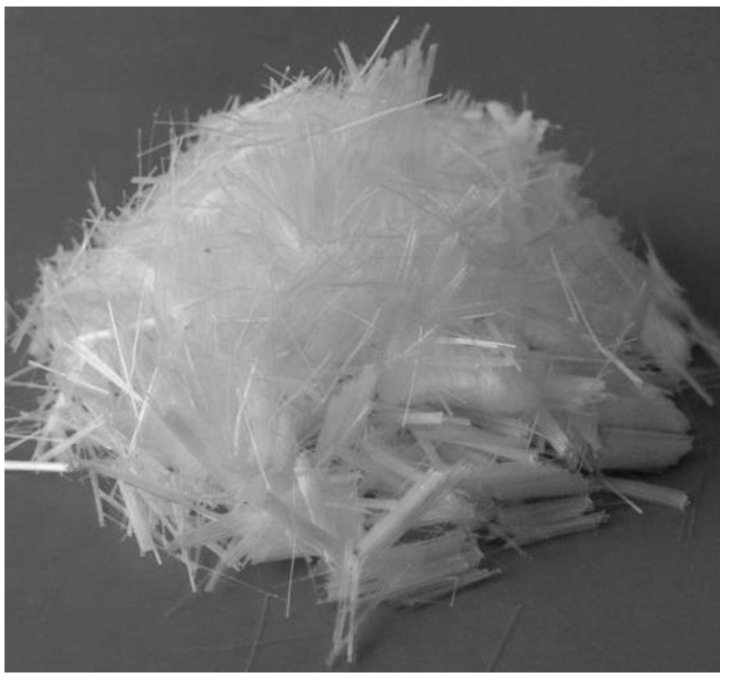
Polypropylene fibers.

**Figure 4 materials-15-07923-f004:**
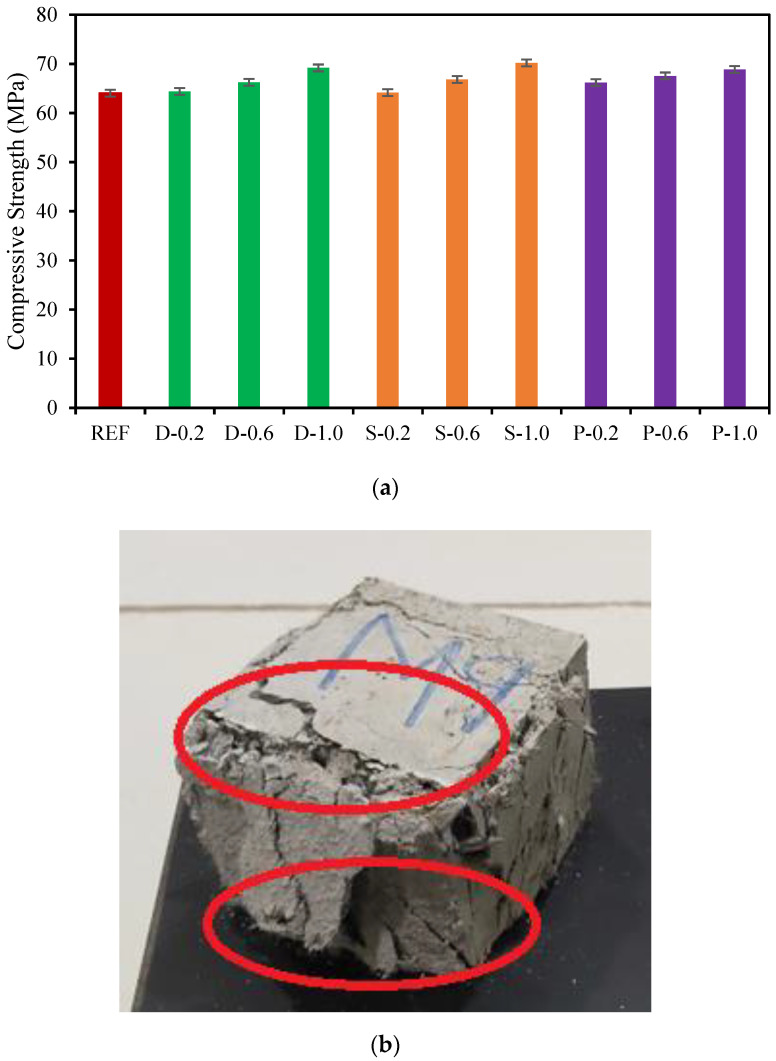
(**a**) Compressive strength of fibrous-reinforced high-strength concrete.; (**b**) Ductile failure of samples in compression. Read circles identify first cracking zones, indicating the ductile behavior.

**Figure 5 materials-15-07923-f005:**
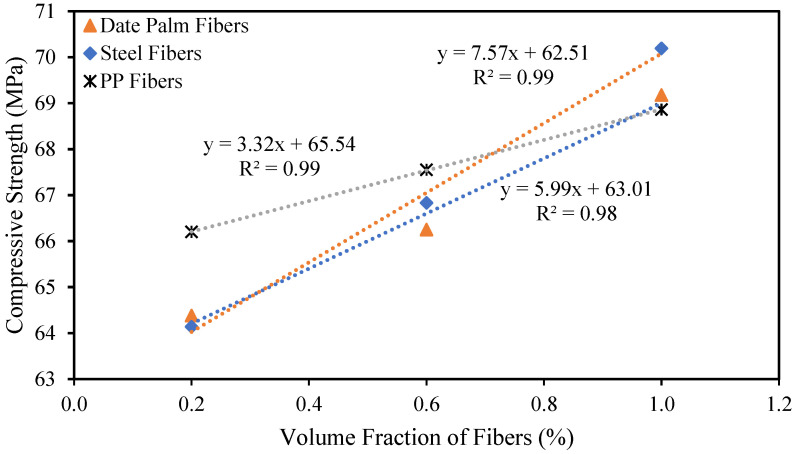
Relationship between the compressive strength and fractions of different fibers.

**Figure 6 materials-15-07923-f006:**
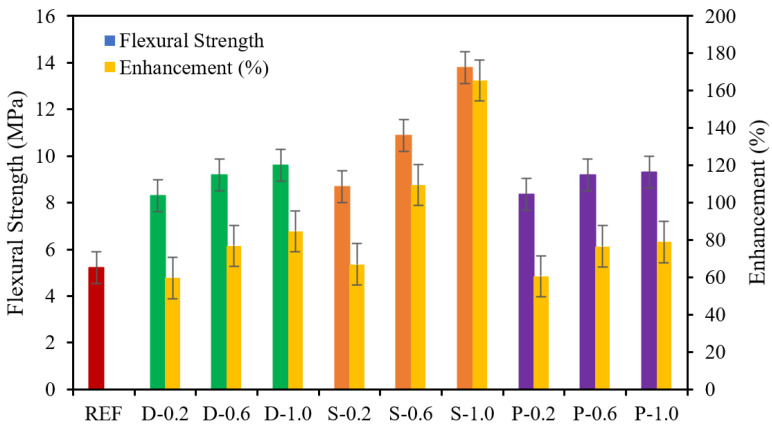
Flexural strength and flexure enhancement of fibrous concrete samples.

**Figure 7 materials-15-07923-f007:**
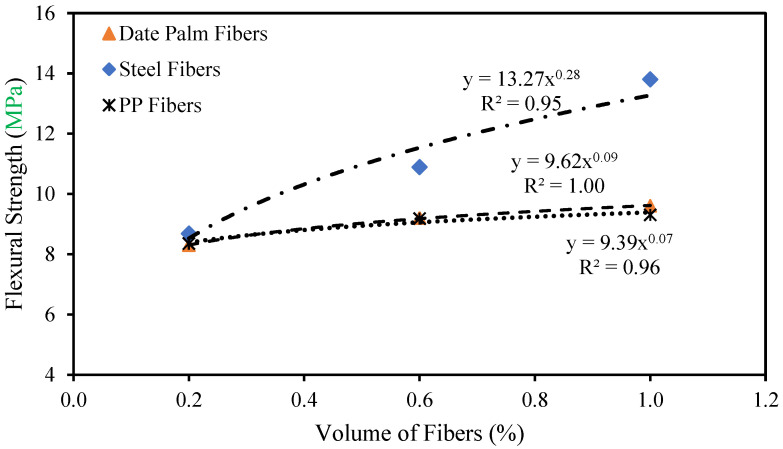
Relationship between the flexural strength and volume fractions of different fibers.

**Figure 8 materials-15-07923-f008:**
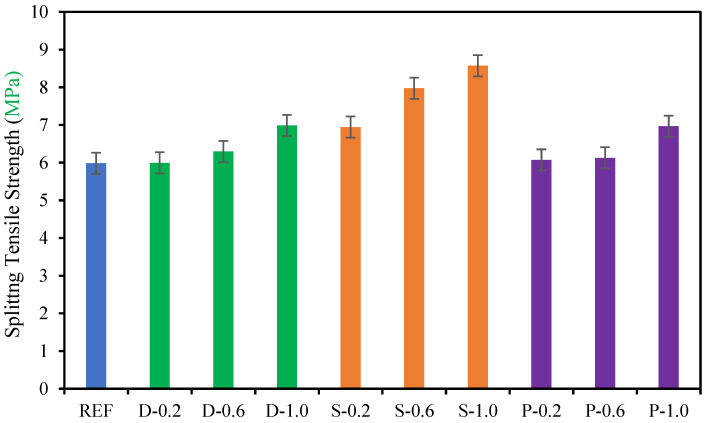
Splitting tensile strength of fibrous concrete samples.

**Figure 9 materials-15-07923-f009:**
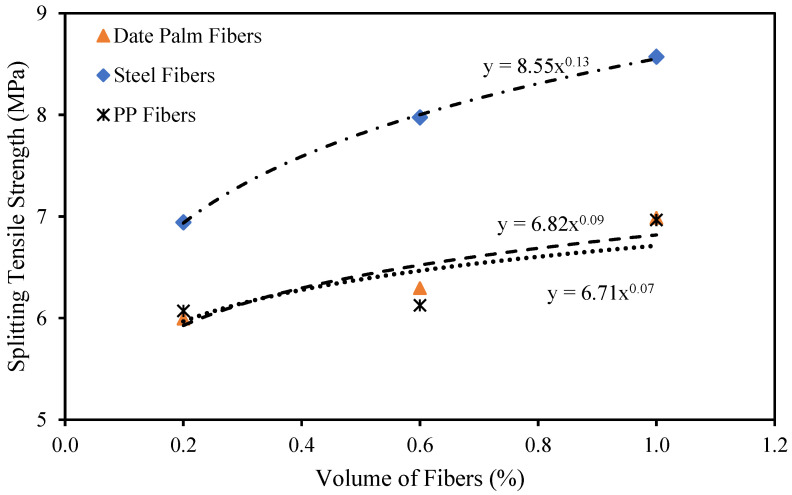
Relationship between the splitting strength and volume fractions for different fibers.

**Figure 10 materials-15-07923-f010:**
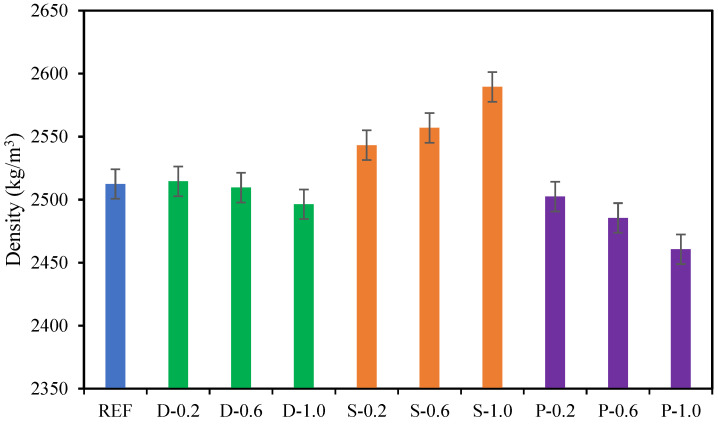
Densities of fibrous concrete samples.

**Figure 11 materials-15-07923-f011:**
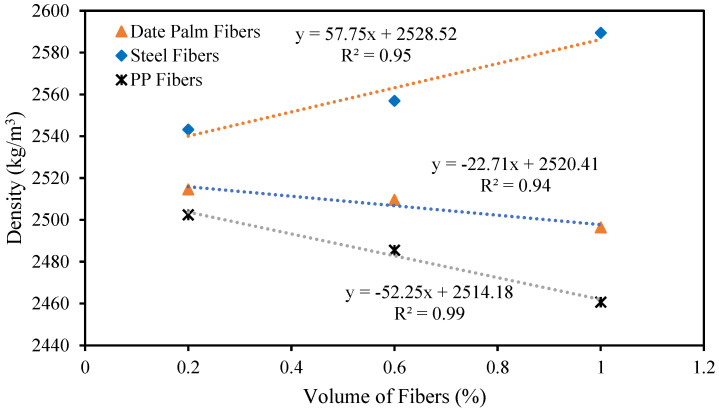
Relationship between the density and volume fractions of different fibers.

**Figure 12 materials-15-07923-f012:**
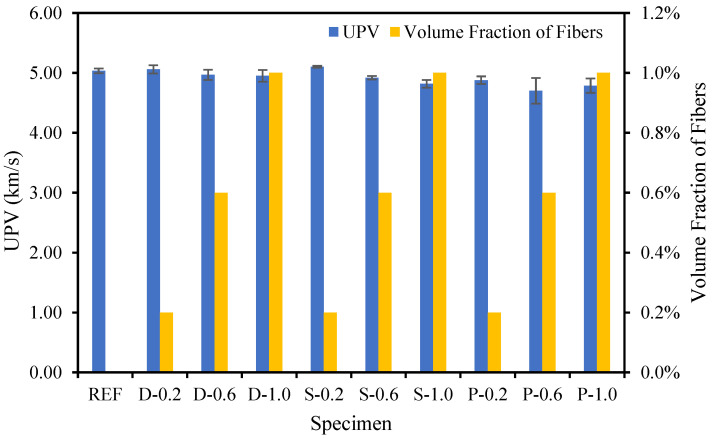
Different fiber influences on the UPV of fibrous concrete samples.

**Figure 13 materials-15-07923-f013:**
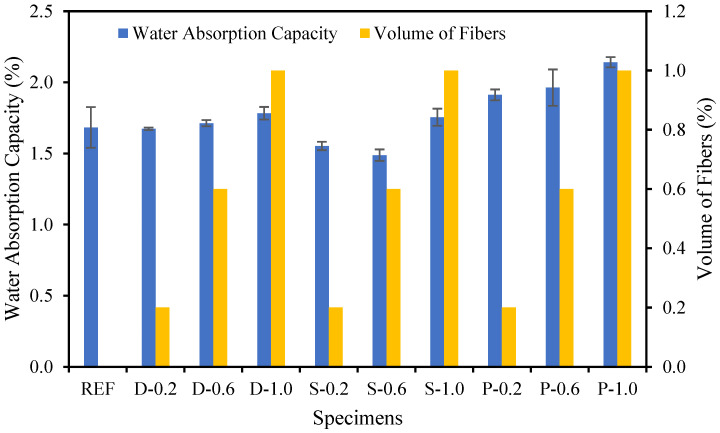
Different fibers influence on the water absorption capacity of fibrous concrete samples.

**Figure 14 materials-15-07923-f014:**
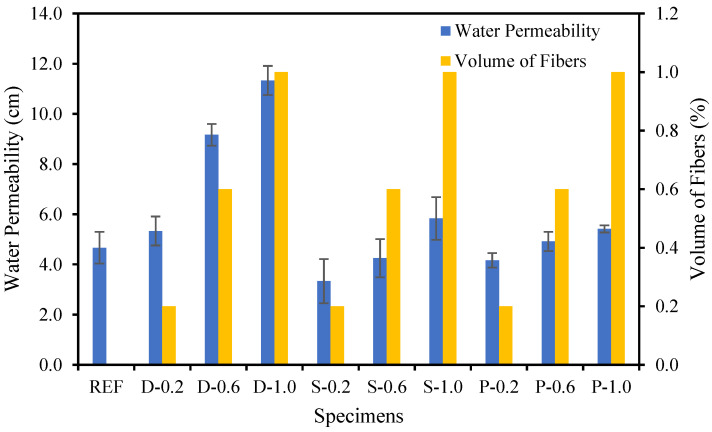
Different fibers influence on the water permeability of fibrous concrete samples.

**Figure 15 materials-15-07923-f015:**
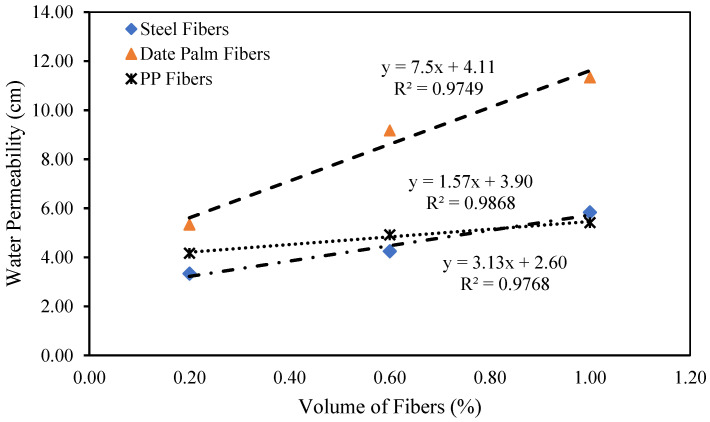
Relationship between the water permeability and volume fractions for different fibers.

**Figure 16 materials-15-07923-f016:**
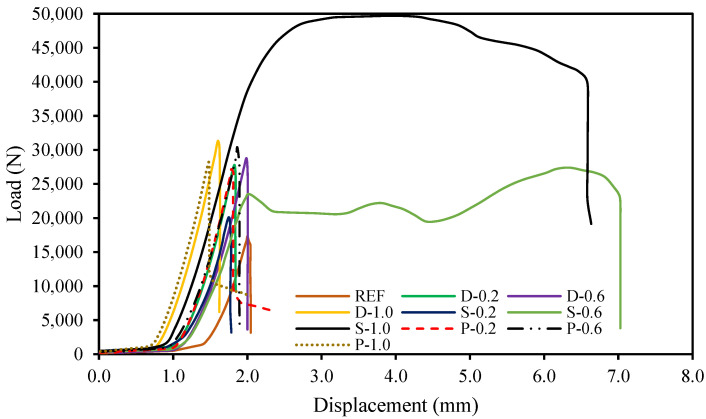
Load-displacement characteristics of the fibrous-reinforced concrete.

**Figure 17 materials-15-07923-f017:**
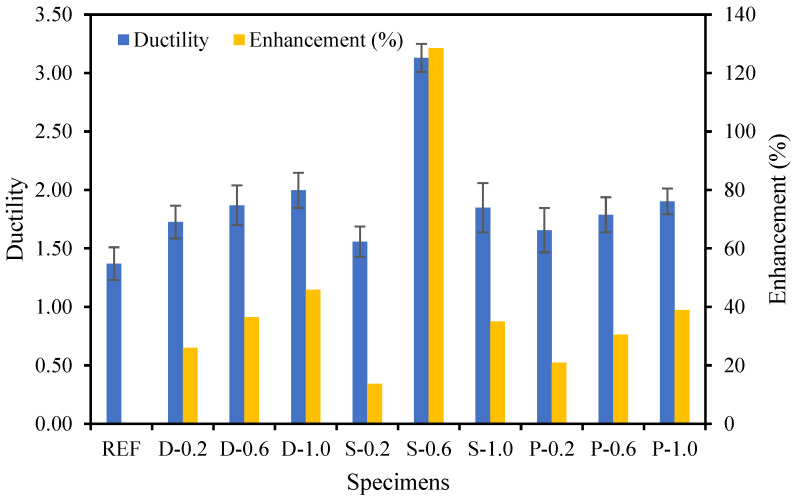
Ductile performance of the fibrous-reinforced concrete.

**Table 1 materials-15-07923-t001:** Cement chemical composition.

Chemical Compound	Cao	SiO_2_	Al_2_O_3_	Fe_2_O_3_	SO_3_	LOI	K_2_O	MgO	Insoluble
**Weight (%)**	63.83	19.70	6.25	3.45	2.25	1.52	1.08	0.97	0.95

**Table 2 materials-15-07923-t002:** Physical characteristics of the aggregates.

Type of Aggregate	Bulk Density (kg/m^3^)	Specific Gravity	Fineness Modulus	Absorption (%)
Fine	1535.74	2.67	2.23	1.31
Coarse	1630.00	2.77	7.34	0.69

**Table 3 materials-15-07923-t003:** Physical properties of date palm fibers [35].

Length (mm)	Diameter (mm)	Density (kg/cm^3^)	Tensile Strength (MPa)	Elongation (%)	Modulus of Elasticity (GPa)
60	0.1–1.0	1.30	240 ± 30	12 ± 2	5.00 ± 2

**Table 4 materials-15-07923-t004:** Physical properties of steel fibers.

Length (mm)	Diameter (mm)	Aspect Ratio	Density (kg/m^3^)	Tensile Strength (MPa)
60	0.75	80	7850	625

**Table 5 materials-15-07923-t005:** Polypropylene fiber properties.

Length (mm)	Diameter (µm)	Density (g/cm^3^)	Young Modulus (GPa)	Elongation at Breaking (%)	Tensile Strength (MPa)
12	25	0.91	5.4	30	550

**Table 6 materials-15-07923-t006:** High-strength fibrous concrete mixtures (kg/m^3^).

Mix ID	Fibers				Aggregates		Cement	Water	SP
	Percentage	Steel	Date Palm	PP	Coarse	Fine			
REF	0	-	-	-	1105.4	736.93	400	176.4	2
D-0.2	0.2	-	8	-					
D-0.6	0.6	-	24	-					
D-1.0	1.0	-	40	-					
S-0.2	0.2	8	-	-					
S-0.6	0.6	24	-	-					
S-1.0	1.0	40	-	-					
P-0.2	0.2	-	-	8					
P-0.6	0.6	-	-	24					
P-1.0	1.0	-	-	40					

## Data Availability

All the data available in the manuscript.

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
