# Peer review of "Behavior of Concrete Reinforced with Date Palm Fibers"

_materials, 2022, doi:10.3390/ma15227923_

Round 1

Reviewer 1 Report

1) Reviewer suggests making 2.5 section separately from Materials section because its preparation and not materials.

2) It would be nice to see more information about the apparatus used for compressive strength determination, i.e. load cell capacity, loading rate, etc. Also, the size of samples, their conditioning conditions prior test, number of samples are missing for all testing methods.

3) Please correct Mpa to MPa in the article.

4) Are you sure it is appropriate to draw regression when only one samples was used for the test of each fiber fraction (Figure 5)? It does not look really reliable.

5) Please remove the green background from the Figure 6a and Figure 6b is not necessary because the bending strength determination method is well known and it is not necessary to show how it is done.

6) Figure 7, Figure 9, Figure 11, etc. The same question. Is it really ok to show a regression through the pints where only one result for ach volume fraction is obtained?

7) Figure 12, Figure 13, Figure 14, Figure 17. there are no upper and lower limits or at least standard deviations of the results presented.

8) I would suggest improving the discussion section with comparison with other authors works, more in depth analysis is required at least in sections 4.1 and 4.2. Moreover, some explanation of the assumed reasons why one or another phenomena is observed are missing. Also, the Introduction lacks to highlight the results of other authors in the same or similar fields. 

9) Reference list formatted not according to the requirements of the journal.

Author Response

Thank you very much for the observation made by the reviewer #1. Consideration of the comments would improve the quality of our manuscript. The paper has been modified based on the reviewer comments.

Reviewer 2 Report

The authors present an interesting paper for the journal Materials. However, the quality of the paper needs to be improved for possible publication. There are some misconceptions and some corrections to be made. My recommendation is Major Revisions.

In the abstract it should be indicated what the fibre percentages used refer to (volume, mass,...).

The literature review in the introduction is very scarce, there are many studies that analyse the behaviour of concretes with different types of vegetable fibres (hemp, coconut, palm, etc.), it is recommended that the authors invest more time in going deeper into the problem. In addition, the objective of the research should be better highlighted.

In the methodology, what are the granulometric curves for the aggregates used, and is information available on their chemical composition?

Why is the use of superpasticizers used in the mixes?

How is the problem of fibre rotting in concrete solved? The process of curing the fibres should be better described, what NaOH proportions have been used?

In Table 6, taking into account that the density of steel is much higher than PPF and Date palm fibre, how is it possible to use the same amount, the dosage in mass does not make sense. The density of the reinforcement materials must be considered.

Figure 5 is obtained by taking only three points to establish the correlation, it does not make much sense with three points to make an adjustment. Clearly this is not a significant volume of data, at least 30 points should be considered to be able to make inferences from these results.

The same comments can be extrapolated to Figure 6, adding in this case that the beam presented does not meet the criterion that between the application of the load and the supports there are more than 2 times the depth of the beam. This is not a test with application to real structures.

The density calculated in 4.4. is apparent or real, this should be clarified. In case it is not real density, is it possible to carry out tests with Helium to know the porosity of the material, these results have a good relation with the capillarity coefficient.

What is the value of Poisson's coefficient used for the calculation of UPV?

Has the standardised water vapour permeability test been carried out?

Finally, with regard to the conclusions, these should be less optimistic and reflect the real limitations of this study and the future lines of work to correct them.

The sections that appear in the original Template of the journal are missing.

Author Response

Thank you very much for the observation made by the reviewer #2. Consideration of the comments would improve the quality of our manuscript. The paper has been modified based on the reviewer comments.

Round 2

Reviewer 1 Report

Authors have taken into consideration all my remarks.

Author Response

Thanks for all the effort and time involved in reviewing our paper, this is greatly appreciated by the scientific community. Based on your comments and subsequent processing, our paper is in a more advanced stage. Thanks.

Reviewer 2 Report

The authors have made all the changes proposed by the reviewer and have significantly improved the quality of the submitted manuscript.

Author Response

(The authors gave the same response as above.)
